# Investigation of the Effects of Adsorbed Water on Adhesion Energy and Nanostructure of Asphalt and Aggregate Surfaces Based on Molecular Dynamics Simulation

**DOI:** 10.3390/polym12102339

**Published:** 2020-10-13

**Authors:** Wentian Cui, Wenke Huang, Bei Hu, Jiawen Xie, Zhicheng Xiao, Xu Cai, Kuanghuai Wu

**Affiliations:** School of Civil Engineering, Guangzhou University, Guangzhou 510006, China; wentiancui1@163.com (W.C.); 13026335960@163.com (B.H.); xjw850843479@163.com (J.X.); xzc297055783@163.com (Z.X.); cx_caixu@163.com (X.C.)

**Keywords:** asphalt-aggregate interface, mineral surfaces, interfacial adhesion, molecular dynamics, moisture effect

## Abstract

The purpose of this study was to investigate the effect of aggregate surface adsorbed water on the adhesive capacity and nanostructure of asphalt-aggregate interfaces at the atomic scale. Molecular dynamics (MD) simulation was performed to measure and analyze the molecular interactions of asphalt binder with calcite and silica. Radial distribution function (RDF) and relative concentration (RC) were applied to characterizing the concentrations and distributions of asphalt components on aggregate surfaces. In addition, debonding energy and adhesion energy were employed to calculate the variations of interface adhesion energy of the asphalt-aggregate system under different conditions. The obtained results illustrated that the water molecules adsorbed onto the surface of weakly alkaline aggregates inhibited the concentration and distribution of asphalt components near the aggregate surface, decreased adhesion energy between asphalt and aggregates, and changed asphalt nanostructure. Especially, when external free water intruded into the interface of the asphalt-calcite system, the adsorbed water interacted with free water and seriously declined the water damage resistance of the asphalt mixture with limestone as an aggregate and decreased the durability of the mixtures. The water adsorbed onto the surface of the acid aggregate negatively affected the asphalt-silica interface system and slightly reduced the water damage resistance of the asphalt mixture.

## 1. Introduction

Asphalt mixtures, which are widely being used in roadway engineering construction, consist of aggregates, mineral powders, and asphalt binders [1]. Adhesive capacity between asphalt and aggregate plays a critical role in the structure of asphalt mixtures and has great impact on the service quality of pavements [2,3,4]. Asphalt pavements are affected by various external factors such as fatigue cracking and rutting, which easily cause microcracks declining the adhesion ability of asphalt mixtures [5,6]. The increase of the interfacial bond at the asphalt-aggregate interface improves the mechanical properties of asphalt mixtures and increases the service life of asphalt pavements.

The existence of moisture in the asphalt mixture decreases bond strength between the asphalt and aggregate, resulting in water damage to the asphalt pavement and inducing other pavement damages [7,8,9]. Liu et al. [10] investigated the interface-cracking performance of asphalt-aggregate systems under the wet condition and found that moisture intrusion caused spontaneous separation of asphalt from the aggregate surface. Jorge et al. [11] studied the effects of water on the asphalt-aggregate adhesiveness of four different asphalt mixtures and proved that water intrusion into the interface resulted in the stripping of asphalt from the aggregate surface. Fauzia et al. [12] developed a new test method to study the combined interaction of the water–tire–pavement. Their results showed that the existence of moisture in the asphalt mixture accelerated the cracking and rutting of the surface. Teh et al. [13] applied a 3-D imaging technique to quantify the percentage adhesive failure of asphalt mixtures and the amount of asphalt stripping from aggregate surfaces under moisture conditions. Therefore, investigation of adhesion failure due to moisture at the asphalt-aggregate interface can reveal the mechanism of water damage in asphalt mixtures and provide effective suggestions for improving the water resistance of asphalt pavements.

Although experimental methods can quantify and evaluate the moisture effect in asphalt mixtures, complex internal structures and components of asphalt binders make it difficult to study the microinteractions of asphalt–aggregate interfaces in macroscopic experiments. Recently, as an emerging technique for the characterization of interactions between substances, molecular dynamics has been adopted by researchers to reveal molecular interactions between aggregates and asphalt binders at atomic scale. Yao et al. [14] developed asphalt–aggregate and aggregate–water models to simulate the mechanism of adhesion energy and proved that the reason for water damage in asphalt mixtures was the difference of adhesion energies between asphalt–aggregate and aggregate–water interfaces. Xu and Wang [15] carried out MD simulations to investigate the adhesive properties of asphalt concretes and found that the cohesive ability between the aggregate and asphalt relied on the type of aggregate mineral under dry and wet conditions. Sun and Wang [16] simulated moisture effects between the asphalt (virgin and aged) and aggregate surface and calculated the adhesion energy of asphalt–aggregate interfaces under different conditions. They found that the asphalt nanostructure was changed by interfacial moisture and water decreased adhesion energy regardless of the aggregate mineral type. Luo et al. [17] performed MD simulations and showed that the anisotropy of aggregate surfaces played a significant role in adhesion energy between asphalt and aggregates. Gao et al. [18] investigated the adhesive properties of bitumen–aggregate interfaces and found that non-bond interaction energy was the main effective factor in adhesion energy between the aggregate and asphalt and the moisture damage resistance of asphalt–aggregate interfaces depended on the chemistry of the mineral surface.

The above studies have mainly investigated the effect of external free water intruded into asphalt–aggregate interfaces on the adhesive capacity and water damage resistance. However, it is difficult to ensure the complete dryness of the aggregate during the preservation process in reality and the insufficient drying time of the aggregate mineral before the preparation of the asphalt mixture can result in the adsorption of water by the aggregate [19,20]. Therefore, in this study, asphalt–aggregate interface systems were developed using the molecular dynamics simulation method to explore the variations of adhesion energy and debonding energy between the asphalt and aggregate under dry and wet conditions.

## 2. Models and Simulation Methods

### 2.1. Asphalt Binder Model

Asphalt binder is a multiscale and complex chemical mixture predominantly consisting of hydrocarbons with small amounts of structurally analogous heterocyclic species and functional groups containing nitrogen, sulfur and oxygen atoms [21]. The asphalt binder model with reasonable structure was established by various methods to characterize the complex chemical properties of asphalt. Claire [22] proposed a mesoscopic model for asphalt and the asphalt mixture, and investigated the triblock-copolymer-modified asphalt mixture. Yao [23] demonstrated three components to describe the asphalt model. The model contained asphaltene, an aromatic, and a saturate (5:27:41), 1,7-dimethylnaphthalene and docosane represented the aromatic and saturate, respectively. Hansen [24] motivated a four-constituent model composed of asphaltene, resin, resinous oil, and saturated hydrocarbon, and the results present that these four components tend to form nanoaggregates. According to Corbett’s [25] analysis on asphalt extraction, the asphalt binder model includes four components: saturates, aromatics, resins, and asphaltenes (SARA). Chu’s [26] constructed asphalt model consisted of asphaltenes, resins, aromatics, and a saturate, and the mass ratio is 1:3:7:5. Li and Greenfield’s [27] established AAA-1 asphalt model constituted of 12 components and this model was adopted in this study, as shown in Figure 1. The detailed component parameters of asphalt models are presented in Table 1.

The MD simulations applied in this research to establish atomic models and calculate thermodynamic properties were performed using commercially available simulation software, Materials Studio (MS, Accelrys, San Diego, CA, USA). The force field that can describe the relationship between molecular potential energy and molecular structure plays an essential role to molecular simulation. To further represent atomic and molecular interactions condensed-phase optimized molecular potentials for the atomistic simulation studies (COMPASS) force field was employed. The parameters of the COMPASS force field were derived from both experimental tests and ab initio methods, and the charge between atoms come from the fitting of quantum chemistry data. Besides, the COMPASS force field has detailed coverage in materials that contain vibrational, conformational, and atomic properties [28,29,30]. Therefore, the COMPASS force field was adopted for calculation. Then, the amorphous cell module in MS was applied to establish the periodic structures of molecular liquids and polymeric systems and the asphalt binder model was also constructed based on Table 1. After that, in the asphalt model, the initial density was set at 0.1 g/cm^3^ and an isothermal-isobaric ensemble (NPT, 298.15 K, 1 atm) with a 1 fs time step was applied for 500 ps to sufficiently relax the simulation system. The Andersen barostat and Nose–Hoover thermostat were used to control constant pressure and temperature, respectively. The model density of the asphalt binder reached a plateau after 300 ps and stabilized at 1.006 g/cm^3^ during the relaxing process, as shown in Figure 2. The validation of the asphalt binder model was evaluated by thermodynamic properties, such as density, cohesive energy density (CED) and solubility parameter. Cohesive energy is the average energy separating all molecules to an infinite distance. CED is a measurement of the intermolecular force, and it can be employed to access the intermolecular interactions in the asphalt binder model. Meanwhile, the solubility parameter (δ), which is the square root of CED, was adopted to estimate the molecular properties. $\delta_{vdw}$ and $\delta_{ele}$ represented the Van Der Waals force and electrostatic interactions between molecules in the asphalt model, and the detailed properties information are shown in Table 2. Asphalt as a viscoelastic material presents glassy behavior at low temperatures and has high viscosity at elevated temperatures [31,32]. The glass transition temperature (T_g_) of the asphalt binder is a characteristic temperature and there exists a reversible change from a brittle glassy state to a rubber-like viscoelastic state or vice versa. When the temperature is lower than T_g_, the stiffness of the asphalt is higher and contributes to the decline of deformation capacity. Therefore, T_g_ is closed relevant to low temperature cracking [33]. The free volume theory was adopted in MD simulations to provide the relationship between temperatures and specific volumes. Then, the NPT ensemble simulation at temperatures from 238 to 338 K was conducted, the changes of a specific volume with temperature are shown in Figure 3. It can be seen from Figure 3 that the specific volume had an increasing trend as temperature rising, and the specific volume increase rate at elevated temperatures was higher than that at low temperatures. Two different regions were fitted respectively, and the intersection point 290.7 K, which is the glass transition temperature of our selected asphalt, was obtained. After that, the COMPASS force field was carried out to further analyze the asphalt binder model. Based on the results reported in the literature, the developed model could be recognized as a valid model for asphalt binders [16,26,34].

### 2.2. Aggregate Mineral Model

Aggregates that are widely employed as construction materials contain limestone, granite, basalt, sand, etc. As a classical component among all aggregate materials used in asphalt mixtures, α-quartz (SiO_2_) and calcite (CaCO_3_) were selected as acidic and alkaline aggregates, respectively [16,36,37]. Silica (α-quartz) and calcite crystals were obtained from the Cambridge Structural Database (CSD, Cambridge Crystallographic Data Centre, London, UK). However, the residual water adsorbed by the aggregate could transform silica to hydrated silica (SiO_2_·nH_2_O). Due to nH_2_O characterized in the form of hydroxyl, a fully-hydroxylated SiO_2_ surface was established to represent the aggregate surface. CO_3_^2−^ and Ca^2+^ ions in calcite are prone to necking of electron density and the ionic bonds between them can easily break [38,39]. CO_3_^2−^ ions can easily form bonds with H+ ions in residual adsorbed water to form HCO_3_^−^ [40,41]. The presence of adsorbed water in alkaline aggregates results in the creation of a layer of HCO_3_^−^ ions on aggregate surface. Therefore, in this study, four different aggregates (silica and calcite in the presence and absence of adsorbed water) were employed as the aggregate layer, as can be seen in Figure 4 and Figure 5.

The unit cell of silica was cleaved to acquire a (0 0 1) surface and the 5 × 5 × 6 supercell structure of silica was established. Then, a thin vacuum slab was added on the top of the extended molecule surface to construct a three-dimensional periodic boundary model. The super unit cell method of calcite was constructed similar to that of silica. After that, the model of silica with adsorbed water was established by adding hydroxyl on the surface of the silica model (silica-hydrated surface model) and that of calcite with adsorbed water (calcite-hydrated surface model) was developed through appending the hydrogen bond to CO_3_^2−^ ions on the surface of the calcite model.

### 2.3. Asphalt–Aggregate Mineral System Model

The models of asphalt and the aggregate mineral were combined to construct asphalt–aggregate interface models through the build layer tool. A 50 Å vacuum layer was added on the top of each asphalt–aggregate interface model to eliminate the effect of the 3D periodic boundary condition. Moreover, for the aggregate surface modification caused by the presence of moisture, the “Asphalt–aggregate-hydrated surface (HS)” models were established to simulate the effects of adsorbed water in the asphalt–aggregate system. After the energy minimization process, the most stable structures were obtained, as shown in Figure 4.

The existence of free moisture at the interface of the asphalt–aggregate system is a significant cause for damage creation in asphalt mixtures. Thus, to investigate the effects on moisture on the water damage resistance of asphalt mixtures, asphalt–water–aggregate models were constructed. Three hundred water molecules [42] were added to the asphalt–aggregate interface to simulate the possible behaviors of moisture intrusion in reality. The optimized models can be seen in Figure 5.

The Anderson thermostat was used to control the temperature of the structure and the COMPASS force field was employed in the following all-atom molecular dynamics simulation. Van der Waals interactions with a cutoff distance of 15.5 Å were based on atom summation. Ewald summation method with a 6 Å cutoff distance was applied for electrostatic interactions. Then, interface models were conducted for 500 ps controlled by constant volume and temperature (NVT) ensemble at 298.15 K. MD simulations were performed with a time step of 1 fs.

### 2.4. Radial Distribution Function

Radial distribution function (RDF) is employed to describe the normalized probability of a particle that occurs around a specific center particle at a given radial distance r in a system [43]. In MD simulations, RDF curves show the variations of density and the morphological structures of components in order to predict changes of other thermodynamic and mechanical properties. Thus, RDF was calculated to characterize microstructure varieties of the interface system due to moisture intrusion into the asphalt–aggregate interface. RDF g(r) is shown in Equations (1) and (2) [44].
(1)$$\rho g\left(r\right)4\pi r^{2}=dN$$
(2)$$\int_{0}^{\infty }\rho g\left(r\right)4\pi r^{2}dr=\int_{0}^{N}dN=N$$
where ρ is the density of interface system, N is the total number of molecules, g(r) is the interval distance, and r is the radial distance.

### 2.5. Interfacial Adhesion Energy and Debonding Energy

Adhesion energy was applied to evaluate the adhesive capacity between asphalt binders and aggregates, which was closely related to the fracture resistance and durability of asphalt mixtures. The adhesion energy of asphalt–aggregate models was derived by Equation (3) [16].
(3)$$W_{adhesion}=∆E_{asphalt-aggregate}=E_{total}-E_{asphalt}-E_{aggregate}$$
where, $W_{adhesion}$ is the adhesion energy between the aggregate and asphalt; $∆E_{asphalt-aggregate}$ is aggregate–asphalt interaction energy; $E_{total}$ is the whole potential energy of the asphalt–aggregate system; and $E_{aggregate}$ and $E_{asphalt}$ are aggregate and asphalt model potential energies, respectively.

The hydrophobic nature of the asphalt binder and hydrophilicity of minerals facilitates the external water intruded into the asphalt–aggregate interface. The water tends to separate the asphalt from the aggregate surface and thus has a negative effect between the aggregate and asphalt binder. Therefore, it is essential to investigate the debonding between asphalt and the mineral in the asphalt mixture under wet conditions. Debonding energy is the work required to replace the asphalt binder in the asphalt–aggregate model by the intruded moisture, as expressed in Equation (4) [16].
(4)$$W_{debonding}=∆E_{asphalt-water}+∆E_{aggregate-water}-∆E_{asphalt-aggregate}$$
where $W_{debonding}$ is the work of debonding; $∆E_{asphalt-water}$ is the bonding energy of the asphalt–water interface; $∆E_{asphalt-aggregate}$ is the bonding energy of the asphalt–aggregate interface; and $∆E_{aggregate-water}$ is the bonding energy of the aggregate–water interface.

## 3. Results and Discussion

### 3.1. Distribution of Asphalt Compositions on Aggregates

The radial distribution functions of SARA components, which can be output from trajectories, were employed to further characterize the variations of molecular structure in asphalt on the aggregate surface. The RDFs of SARA components were conducted on the analysis of the last 200 ps of NVT simulations and the obtained results are shown in Figure 6a,b and Figure 7a,b.

Figure 6a shows that in asphalt–calcite interface systems, compared with the case with no adsorbed water on the calcite surface, in the presence of water the distributions of resin and aromatic compounds firstly accumulated and the value of g (r) was moved from 0 to 0.3 at the position of 2.1 Å. This was due to electrostatic attractions between hydrogen bonds on the surface of weakly alkaline minerals and the polarity of resin. However, the peak values of all asphalt components were decreased and the distribution of SARA became more uniform, which meant that the water adsorbed on the calcite surface affected the adhesive behavior of the asphalt binder on alkaline minerals. Figure 6b presents that in asphalt–silica interface system in dry conditions, the peak values of the non-polar lightly saturate and resin with strong polarity were close at 16 Å and the distributions of SARA components remained similar after 23 Å, indicating that the interactions between acidic minerals and asphalt were weak and it was hard to change the aggregation state of the asphalt binder around acidic minerals. When water was adsorbed onto the surface of silica, the charge effect of surface hydroxy could easily attract the resin, which was firstly adsorbed at 3 Å and the peak value also was increased slightly at 13 Å. Then, the distributions of SARA components near silica surface containing adsorbed water were the same trend as that of silica without adsorbed water after 23 Å. The obtained results showed that the presence of adsorbed water on the surface of acid aggregates affected the distributions of asphalt components.

When external free water intruded into the interface of the asphalt–aggregate system, it exerted different effects on asphalt–calcite and asphalt–silica models. In Figure 7a, asphaltene reached the first peak at 15 Å under the influence of interfacial moisture when there was no adsorbed water on the surface of calcite and this showed that asphaltene inclined to aggregate on weakly alkaline mineral surface. However, the distribution of asphaltene was more uniform in the presence of adsorbed water on the calcite surface. This was not beneficial for the interaction of calcite and asphaltene and the adhesive ability between asphalt and calcite was attenuated. Comparison of Figure 6b and Figure 7b obviously shows that the existence of free water at the asphalt–silica interface delayed the g(r) peak of the resin from the initial 13 to 20 Å, expanded the distance between asphalt and SiO_2_, and declined the adhesion energy of the asphalt–silica system. Figure 7b also shows that the existence of interfacial moisture did not affect the distributions of SARA components on the silica surface.

Concentration distribution along vertical direction from the interface can further estimate the concentration values near the aggregate surface. Dynamic trajectory was employed to evaluate relative concentration along the X axis, as shown in Figure 8a,b and Figure 9a,b.

The water adsorbed on the surface of calcite had a distinct impact on the asphalt–calcite interface system, as can be seen in Figure 8a. Adsorbed water significantly decreased the peak concentration of the resin, extended the appearance distance of the resin and asphaltene, and declined the aggregation concentrations of substances with high polarity (asphaltene and resin) on the surface of the calcite. Due to lower mass fractions of saturate components, the diffusion rate of the saturate was increased and its concentration was reduced under the effect of adsorbed water on the calcite surface. Thus, it was seen that the water adsorbed on the surface of alkaline minerals decreased adhesion energy between the asphalt binder and aggregate through inhibiting the concentration of polar substances in asphalt on the aggregate surface. Figure 8b shows that in the asphalt–silica interface system, the peak concentration of asphaltene was basically unchanged even if water was adsorbed on the silica surface. The first peak of the saturate was shifted from 4.5 to 8 Å, while the peak concentration still remained constant. These results indicated that the water adsorbed on the surface of acid aggregate increased the distance between the asphalt and aggregate, thus affecting the adhesive ability of the asphalt–silica interface.

As can be seen in Figure 8a and Figure 9a, under the effect of external free moisture at the asphalt–calcite interface, the peak concentrations of asphaltenes and resins on the calcite surface significantly declined and the concentration distribution became more uniform. Then, when the calcite surface adsorbed water, the concentration peak of asphaltene and resin was seriously shifted (16 Å for asphaltene and 7.5 Å for resin) with the decrease of the concentrations of asphaltene and resin. These results illustrated that the water adsorbed on the calcite surface and free water at the interface could work together to reduce the concentration of polar molecules in asphalt on weakly alkaline aggregate surfaces. Figure 9b exhibits that when interfacial water and silica surface adsorbed water coexisted, the peak value of the resin was delayed by 5 Å and the peak concentration of asphaltene was slightly decreased, which enhanced adhesion failure between asphalt and the acid aggregate.

The results obtained for the distributions and concentrations of SARA components near aggregates showed that when water was adsorbed on the surface of weakly alkaline and acid aggregates, the adsorbed water had the greatest negative impact on adhesion energy between asphalt and calcite. Furthermore, the intrusion of external free water onto the interface of the asphalt–aggregate system was more harmful than adsorbed water to the adhesive ability of the asphalt–aggregate system.

### 3.2. Adhesion Energy Between the Aggregate and Asphalt

Adhesion energy between asphalt and the aggregate is a significant factor in the determination of the durability and sustainability of asphalt mixtures. The MD simulation method can effectively evaluate the impact of moisture on the adhesion energy of the asphalt–aggregate interface system under different conditions. The variations of the adhesion energy of asphalt with weakly alkaline aggregate calcite and acid aggregate silica were simulated under different conditions such as the dry aggregate, adsorbed water on the aggregate surface, and free water at the asphalt–aggregate interface. The water damage resistance of the asphalt–aggregate system was also analyzed.

The adhesion energies of the asphalt–aggregate system are illustrated in Figure 10. For the asphalt–calcite system, adhesion energy was 467.81 kcal/mol under dry condition. When calcite surface adsorbed water or there was the external free water at the interface, the adhesion energies of the system became 453.26 kcal/mol and 416.51 kcal/mol and energy reductions were 3.1% and 11.0% compared with the dry state, respectively. The obtained results showed that the presence of adsorbed water on the surface of weakly alkaline aggregates reduced the adhesion energy of the asphalt–calcite interface and free water at the interface decreased adhesion energy even more significantly. When there were both free external water in the asphalt calcite system and adsorbed water on the aggregate surface, adhesion energy between asphalt and calcite declined the most and adhesion energy was 390.19 kcal/mol, which was decreased by 16.6%. It could be observed that the combined action of adsorbed water and free water further reduced the adhesive capacity of the asphalt–calcite system and affected the service life of asphalt mixtures [45,46]. For the asphalt–silica system, adhesion energy was 103.53 kcal/mol under the dry condition. When silica surface adsorbed water, the adhesive ability between asphalt and acidic aggregate was slightly improved and the adhesion energy was 115.41 kcal/mol. The reason for this was that although adsorbed moisture increased distance between asphalt and aggregate, the fully hydroxylated silica surface with powerful polarizability could attract polar substances in asphalt and increase the adhesion energy of the asphalt–silica system. After external free water intruded into the interface of the asphalt–silica system, the adhesion energies of silica surface with and without adsorbed water were 28.17 kcal/mol and 28.40 kcal/mol, respectively, which showed a small difference. The adhesion energy of the asphalt–silica system under the action of external free water was reduced by 72.8% compared with that in the dry state, which seriously decreased the durability of the asphalt mixture.

As illustrated in Table 3, the debonding energies of different asphalt–aggregate interface systems were negative. Negative debonding energy showed that external free water at the interface could peel the asphalt binder from the aggregate surface without requiring external energy, indicating that a separation process occurred spontaneously. The absolute value of debonding work reflects the moisture damage resistance of the asphalt–aggregate interface system. The debonding works of asphalt–calcite, asphalt–calcite–HS, asphalt–silica, and asphalt–silica–HS systems were −6649.36, −4974.34, −50.25, and −42.13 kcal/mol, respectively. The weakly alkaline aggregate had the highest resistance to water damage. However, due to the presence of adsorbed water on the surfaces of calcite and silica, the debonding energy of asphalt–calcite and asphalt–silica systems were decreased by 25.2% and 16.1%, respectively. This indicated that, the moisture adsorbed on the aggregate surface declined the debonding work of the asphalt–aggregate interface system and had the greatest impact on the adhesive capacity between the weakly alkaline aggregate and asphalt.

## 4. Conclusions

In this study, MD simulations were employed to investigate the variations of the concentration of asphalt components and the effects of the adhesion ability between asphalt and the aggregate under different conditions due to the presence of adsorbed water on the surface of aggregates in asphalt–aggregate interface systems. The following simulation results and conclusions were drawn:

(1) In the asphalt–calcite interface, the water adsorbed on the surface of weakly alkaline aggregates significantly decreased the aggregation concentrations of resin and asphaltene and made the distributions of SARA components near the calcite surface more uniform, which seriously affected adhesion energy between asphalt and calcite.

(2) In the asphalt–silica interface, the water adsorbed on the acid aggregate surface increased the distance between asphalt and the aggregate. Then, since surface hydroxy could easily attract resin in asphalt, it alleviated the negative effects of adsorbed water on the adhesive ability between asphalt and silica.

(3) When both external free water and adsorbed water existed at the interface of the asphalt–calcite system, the absolute value of the debonding energy of the system was decreased by 1675.02 kcal/mol compared with the case containing only external free water intrusion into the interface. The water damage resistance of the asphalt–calcite model was greatly decreased. However, the decrease of adhesion energy and debonding energy of the asphalt–silica interface system was mainly caused by interfacial free moisture.

(4) In addition, it was necessary to ensure that aggregates (especially weakly alkaline limestone) were under dry conditions and completely dry the aggregates before mixing asphalt mixture to avoid the adsorption of water onto aggregate surface, which could create negative impacts on the durability and service life of asphalt mixtures.

## Figures and Tables

**Figure 1 polymers-12-02339-f001:**
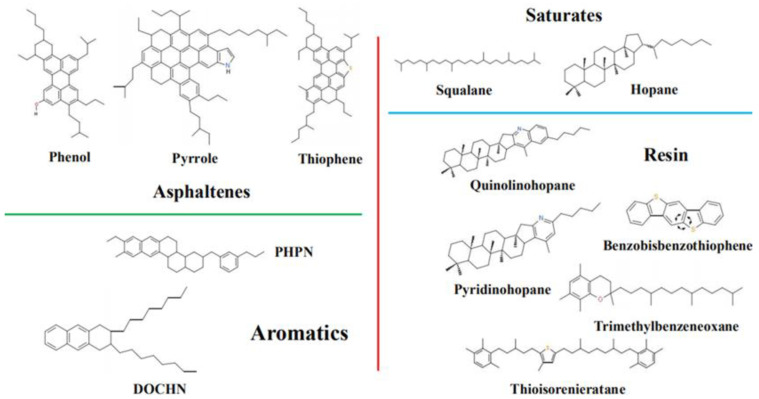
Chemical structures of 12 components in asphalt models.

**Figure 2 polymers-12-02339-f002:**
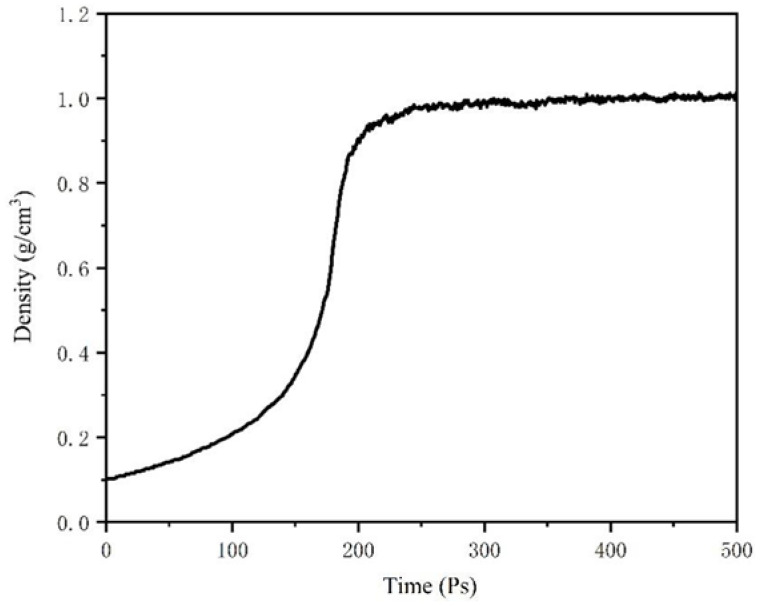
Density curve of asphalt model at 298.15 K in NPT simulation.

**Figure 3 polymers-12-02339-f003:**
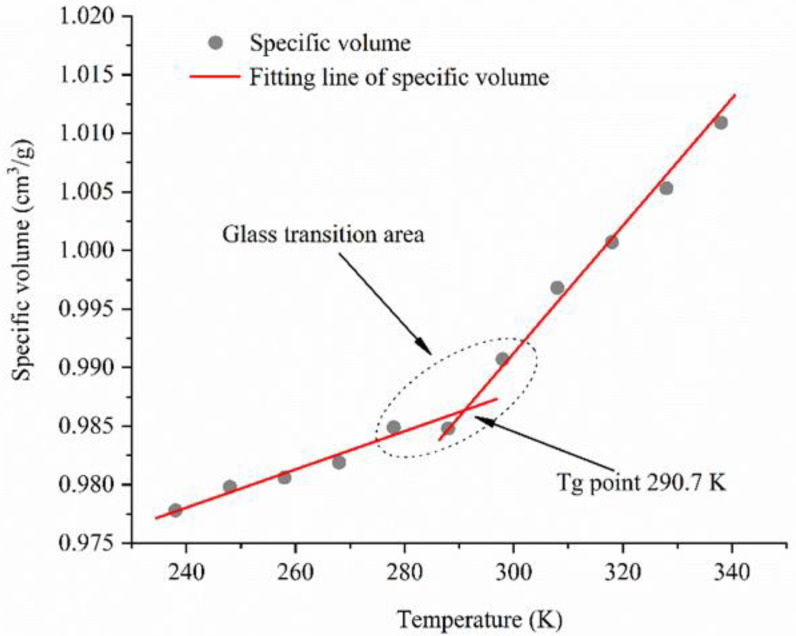
Relationship between specific volumes and temperatures.

**Figure 4 polymers-12-02339-f004:**
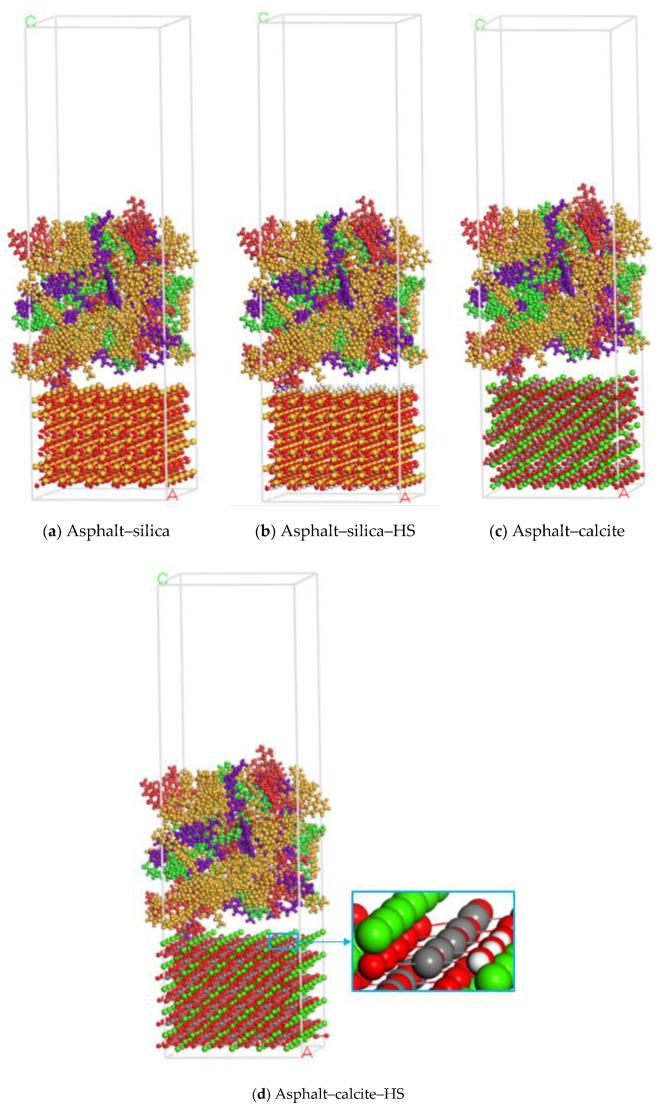
Asphalt–aggregate interface model.

**Figure 5 polymers-12-02339-f005:**
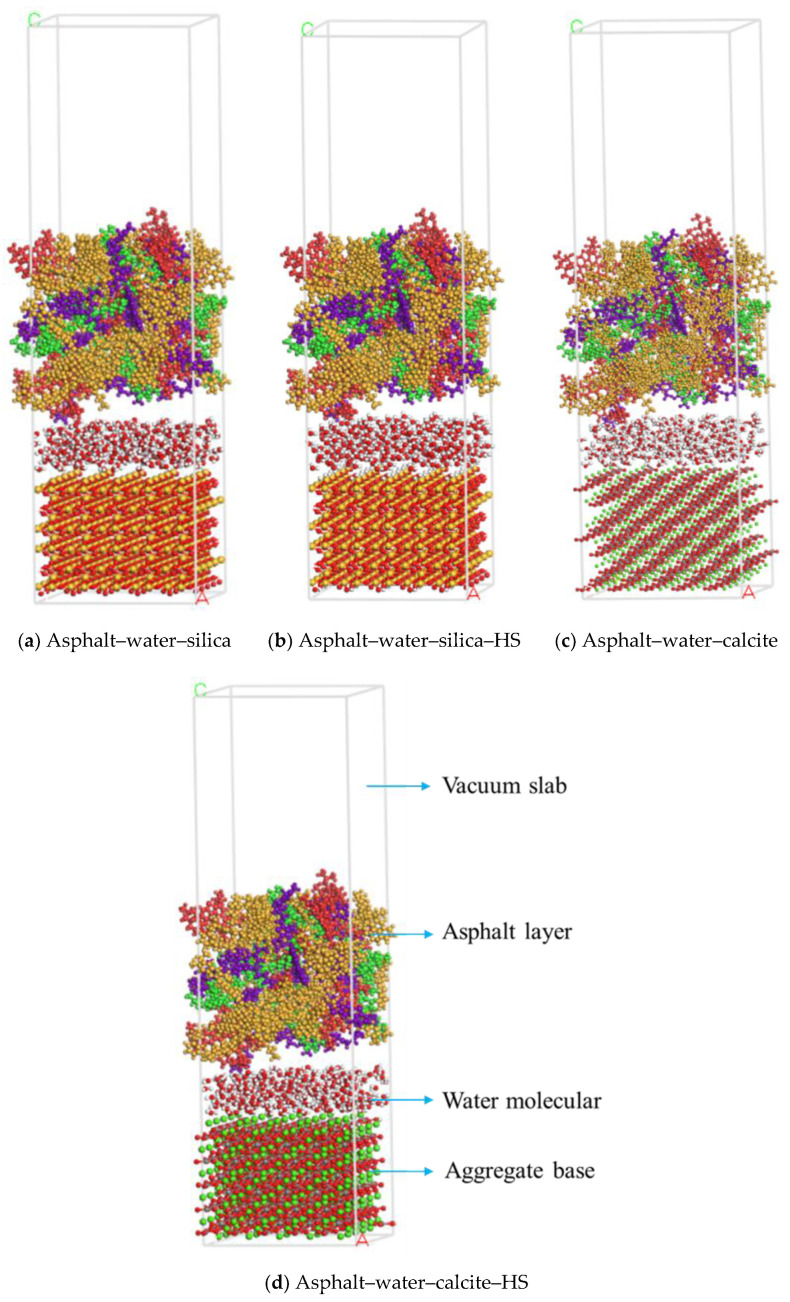
Asphalt–water–aggregate interface model (for asphalt layer, asphaltene: red, resin: purple, saturate: green, aromatic: yellow; for water molecular and aggregate base, carbon: grey, calcium: green, oxygen: red, silica: orange, hydrogen: white).

**Figure 6 polymers-12-02339-f006:**
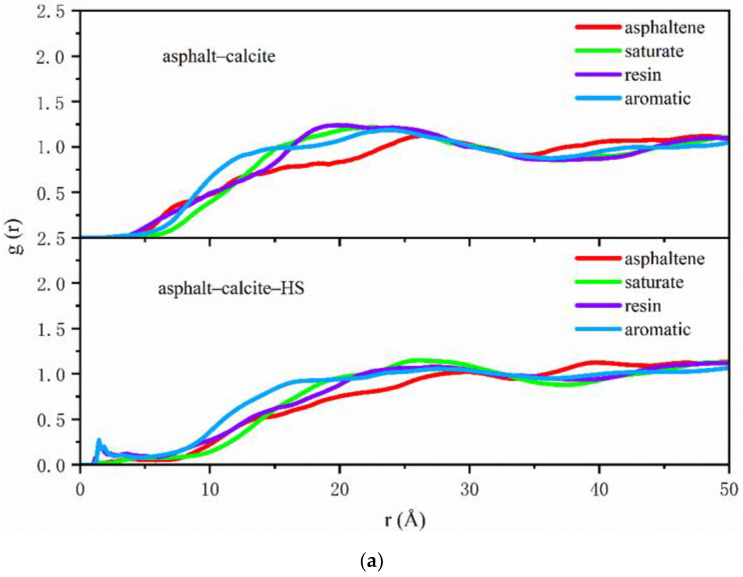
Asphalt–aggregate interface radial distribution function (RDF) curves with and without adsorbed water on the aggregate surface: (**a**) asphalt–calcite and (**b**) asphalt–silica.

**Figure 7 polymers-12-02339-f007:**
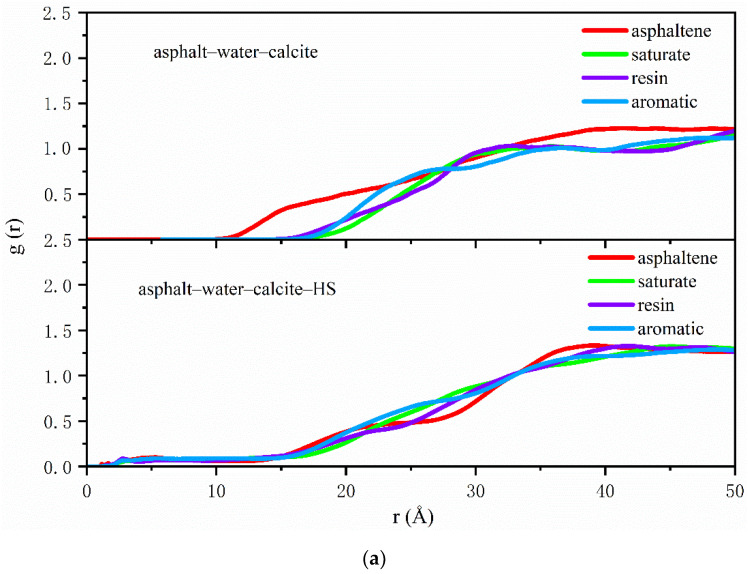
Asphalt–aggregate interface RDF curves under wet conditions with and without adsorbed water on the aggregate surface: (**a**) asphalt–water–calcite and (**b**) asphalt–water–silica.

**Figure 8 polymers-12-02339-f008:**
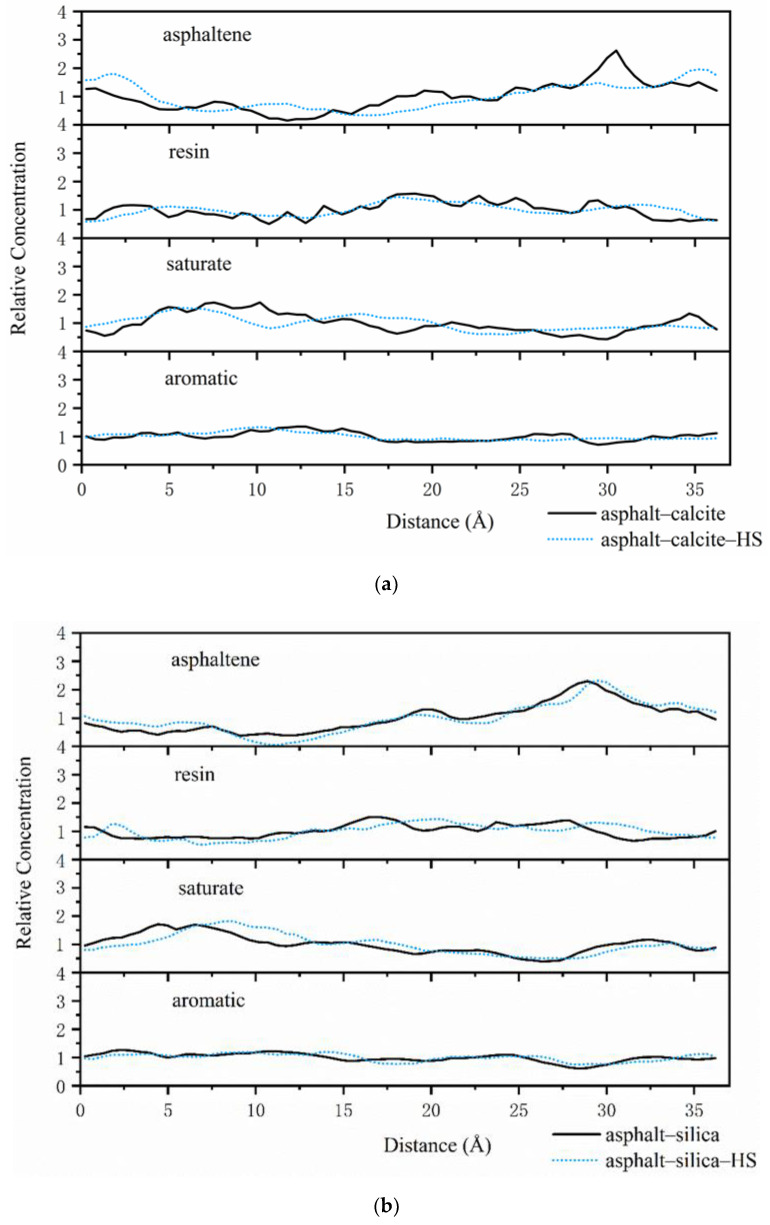
Asphalt–aggregate interface relative concentrations with and without adsorbed water on the aggregate surface: (**a**) asphalt–calcite and (**b**) asphalt–silica.

**Figure 9 polymers-12-02339-f009:**
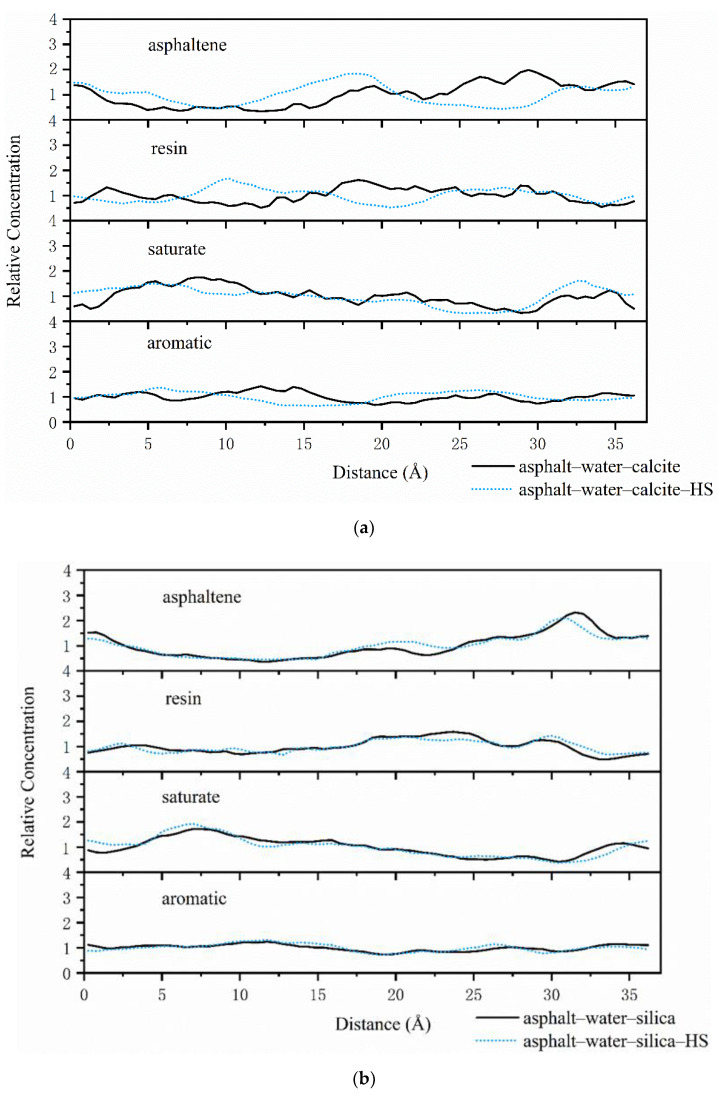
Asphalt–aggregate interface relative concentrations under wet conditions with and without adsorbed water on the aggregate surface: (**a**) asphalt–water–calcite and (**b**) asphalt–water–silica.

**Figure 10 polymers-12-02339-f010:**
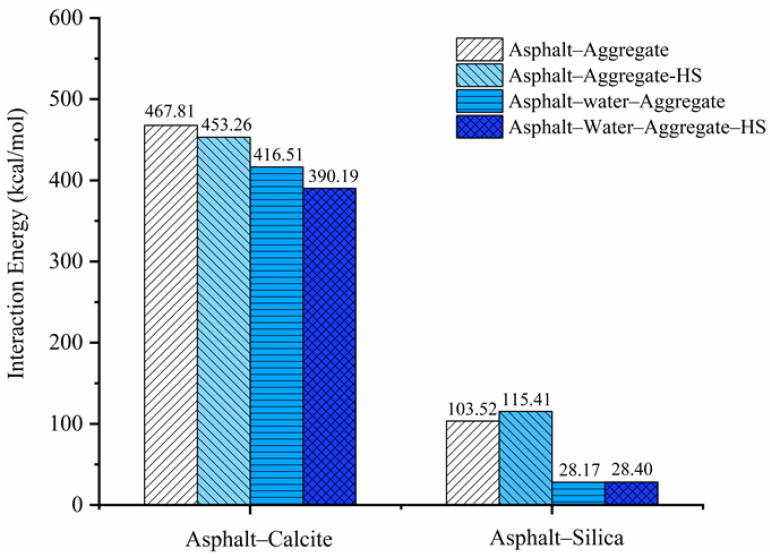
Adhesion energy of the asphalt–aggregate interface system under different conditions.

**Table 1 polymers-12-02339-t001:** Detailed compositions of saturates, aromatics, resins, and asphaltenes (SARA) components in the asphalt model.

Chemical Fractions	Molecules	Formula	Number	Mass Fraction (%)
Asphaltene	Phenol	C_42_H_54_O	3	17.1
	Pyrrole	C_66_H_81_N	2	
	Thiophene	C_51_H_62_S	2	
Saturate	Squalane	C_30_H_62_	4	16.0
	Hopane	C_35_H_62_	6	
Aromatic	PHPN	C_35_H_44_	13	43.6
	DOCHN	C_30_H_46_	16	
Resin	Quinolinohopane	C_40_H_59_N	2	23.3
	Thioisorenieratane	C_40_H_60_S	2	
	Benzobisbenzothiophene	C_18_H_10_S_2_	9	
	Pyridinohopane	C_36_H_57_N	2	
	Trimethylbenzeneoxane	C_29_H_50_O	2	

**Table 2 polymers-12-02339-t002:** Thermodynamic properties of the asphalt model.

Thermodynamic Properties	Simulation Calculation	Experimental Measurements [35]
Density (298.15 K, g/cm^3^)	1.006	1.01–1.04
Cohesive energy density (10^8^ J/m^3^)	3.19	3.19–3.32
δ((J/m^3^)^1/2^)	17.84	13.30–22.50
$\delta_{vdw}$((J/m^3^)^1/2^)	17.321	N/A
$\delta_{ele}$((J/m^3^)^1/2^)	1.496	N/A

**Table 3 polymers-12-02339-t003:** Debonding energy and interfacial interaction for different models.

Models	$∆E_{asphalt-water}$ (kcal/mol)	$∆E_{aggregate-water}$ (kcal/mol)	$∆E_{asphalt-aggregate}$ (kcal/mol)	$W_{debonding}$ (kcal/mol)
Asphalt–Calcite	7531.17	−184,317.15	−170,136.62	−6649.36
Asphalt–Calcite–HS	7825.16	−169,761.52	−156,962.02	−4974.34
Asphalt–Silica	6188.74	−83,958.59	−77,719.60	−50.25
Asphalt–Silica–HS	6265.63	−72,830.21	−66,522.45	−42.13

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
