# Peer review of "Investigation of the Effects of Adsorbed Water on Adhesion Energy and Nanostructure of Asphalt and Aggregate Surfaces Based on Molecular Dynamics Simulation"

_polymers, 2020, doi:10.3390/polym12102339_

Round 1

Reviewer 1 Report

The effect of aggregate surface adsorbed water on the adhesive capacity and nanostructure of asphalt-aggregate interfaces is investigated.  Several important papers in the field of MD simulations of asphalt have been missing.  Authors used a force field that overestimates the density of asphalt in simulations (see comment #3).  The manuscript needs to be revised significantly by adding relevant prior studies in the introduction section.  In addition, the discussion of the density value for the pure asphalt must be included.

  1. Authors use the Li and Greenfield model for asphalt, but it is not clear that why the newer model based on this 12-component system using the GAFF force field has not been used.The force-law of used in the Li and Greenfield model does not show a clear glass transition of asphalt and underestimates the viscosity value.

The modified force field provides properties of asphalt more accurately.  The authors must do a thorough literature review on the model and force field of asphalt.  In addition to this model, the four-component model has also provided reasonable properties for asphalt in both bulk and interface.  A paragraph in the introduction section must be added and overview these recent models of asphalt. 

Important references to be considered in the introduction and comparison of the accuracy of the force field:

(1) J. Phys. Chem. B 2015, 119, 44, 14261–14269

(2) J. Chem. Phys. 149, 214901 (2018)

(3) J. Chem. Phys. 138, 094508 (2013)

(4) Journal of Rheology 62, 941 (2018)

  1. I suggest that authors be cautious with the selection of words in their sentences. e.g.,

“Thus, it can accurately predict material properties.” in line 100.  What does make this force filed to predict material properties accurately? This sentence is too general.  I recommend rewording or deleting it. 

  1. Asphalt is a glass-forming material. I suggest the authors study prior studies on the glass transition of asphalt (for the same 12 component model) in the simulations. The density of the model structure of asphalt at room temperature must be smaller than the experimental density, if the force field “accurately predicts material properties”.  My question is then why the density is exactly in agreement with the experiment?
  2. The manuscript requires a thorough revision from a grammar standpoint.e.g., “Thus, it can accurately predict material properties.” in line 100.

Author Response

Dear Ms. Evelyn Chen and Reviewers,

Thanks very much for taking your time to review this manuscript. I really appreciate all your comments and suggestions! Please find my itemized responses in below and my revisions in the re-submitted files.

Sincerely,

Wentian Cui

Response to Reviewer 1 Comments:

Point 1:Authors use the Li and Greenfield model for asphalt, but it is not clear that why the newer model based on this 12-component system using the GAFF force field has not been used. The force-law of used in the Li and Greenfield model does not show a clear glass transition of asphalt and underestimates the viscosity value. The modified force field provides properties of asphalt more accurately. The authors must do a thorough literature review on the model and force field of asphalt. In addition to this model, the four-component model has also provided reasonable properties for asphalt in both bulk and interface. A paragraph in the introduction section must be added and overview these recent models of asphalt. Important references to be considered in the introduction and comparison of the accuracy of the force field:

(1) J. Phys. Chem. B 2015, 119, 44, 14261–14269

(2) J. Chem. Phys. 149, 214901 (2018)

(3) J. Chem. Phys. 138, 094508 (2013)

(4) Journal of Rheology 62, 941 (2018)

Response 1: We are very sorry for the simple introduction to the asphalt models and molecular force fields, thank you for your important references. We have been carefully studied the articles, and made detailed supplement in 2.1. In addition, 12 components of asphalt binder proposed by Li and Greenfield was used in our study, but the force field we adopted is COMPASS force field. The cohesive energy density (CED) and glass transition temperature (Tg) of asphalt are introduced and the results are presented in Line 128-147, Page 4. Because the focus of our article is to investigate the effect of moisture at the interface of asphalt-aggregate system, we put the introduction of asphalt model in 2.1 (Line 89-100, Page2-3). The detailed description of COMPASS force field can be seen in Line 110-117, Page 3. Thank you again for your valuable suggestions.

Point 2:I suggest that authors be cautious with the selection of words in their sentences. e.g.,

“Thus, it can accurately predict material properties.” in line 100. What does make this force filed to predict material properties accurately? This sentence is too general. I recommend rewording or deleting it.

Response 2: We are very appreciated with your suggestions. We should be more cautious with the selection of words in our sentences. The sentence has been deleted and the introduction of force field has been reworded (Line 110-117, Page 3).

Point 3:Asphalt is a glass-forming material. I suggest the authors study prior studies on the glass transition of asphalt (for the same 12 component model) in the simulations. The density of the model structure of asphalt at room temperature must be smaller than the experimental density, if the force field “accurately predicts material properties”. My question is then why the density is exactly in agreement with the experiment?

Response 3: Thank you for your suggestions. Because the asphalt is a glass-forming material, it is necessary to study the glass transition temperature (Tg) of asphalt model. We have been added these important information (Line 134-147, Page4) and the Figure 3 shows the relationship between specific volumes and temperatures (Line153, Page5). Meanwhile, thermodynamic properties of asphalt model are shown in Table 2 (Line 152, Page 4-5). We approve of your opinion, that sentence is general, and we have been deleted it.

Point 4:The manuscript requires a thorough revision from a grammar standpoint.e.g., “Thus, it can accurately predict material properties.” in line 100.

Response 4: Thank you for pointing out the shortcomings of this article. We have been reviewed the manuscript from a grammatical perspective. We appreciated your help, the references allowed us to better understand the properties of asphalt.

Reviewer 2 Report

The manuscript “Investigation of the Effects of Adsorbed Water on Adhesion Energy and Nanostructure of Asphalt and Aggregate Surfaces Based on Molecular Dynamics Simulation” aims to describe the effect of water adsorbed on the surface of aggregate interfaces on the adhesive capacity and the nanostructure of asphalt/aggregate interfaces.

This work followed very closely the work of Sun and Wang (2020), with a similar choice of models and a similar way of discussing their results. The novelty of this manuscript would then be the modification of the aggregate surfaces by the adsorbed water. In the case of the silica aggregate, with the presence of terminal OH- groups at the surface, while in the case of the calcite aggregate, with the introduction of HCO3- ions adsorbed to the surface. This would be an interest follow-up on the discussion of the interactions between macroscopic organic molecules and inorganic surfaces.

However, there are several main points that should be attended in the manuscript:

1. Still on a conceptual level, the authors did not precise if their model had an overall charge. Besides, the representation of the water-modified surfaces, identified in the asphalt/calcite aggregates with the suffix “-AW”, was rather confusing, with little detailing of the specific modification with the introduction of OH- and HCO3- ions. This “adsorbed water” notation was indeed misleading, since it induces the reader to think there is actually water adsorbed at the interface when it is just a surface modification caused by the presence of water. I suggest changing the “adsorbed water (AW)” notation to another less confusing term, e.g. “modified surface (MS)” or “hydrated surface (HS)”, to avoid this ambiguity. This applies to the whole of the manuscript, the term “adsorbed water” should be used only for the actual interfacial water simulated in the “aggregate-water-asphalt” models.

2. Regarding the construction of the asphalt-aggregate models with adsorbed water at the interface, the authors did not precise the amount of water used in their models and neither justified such amount with experimental data or other previously simulation results.

3. The authors also did not precise their choice of charges. Even though the COMPASS forcefield, as implemented in the Materials Studio suite has built-in charges, this should be explicitly cited in the text.

4. The authors should precise in further details the debonding energy in section 2.5.

5. In their results and discussion section, the authors draw some conclusions that need further validation, such as:

- In line 192-193, “This was due to electrostatic attractions between hydrogen bonds on the surface of weakly alkaline minerals and the polarity of resin”. A RDF between the polar groups of the resin and the proximal aggregate atoms should be presented to support this affirmation.

- In line 211-212, “This was not beneficial This was not beneficial for the interaction of calcite and asphaltene and the adhesive energy of asphalt-calcite interface was attenuated.”. A mention to the calculated adhesive energies in section 3.2 should have been stated.

- In line 291 “(…) and affected the service life of asphalt mixtures.” A reference should be mentioned to relate this cause-effect relationship.

6. The choice of color for the RDFs is very confusing, specially the yellow curves in a white background. It is quite difficult to distinguish some of these curves in some cases.

7. In lines 230 and 231, the term “Dynamic trajectory was employed to evaluate relative concentration along X direction” was confusing. The authors tracked the relative concentration along the vertical direction and should not mention “X direction” but “X axis”.

8. The authors should be homogeneous on their terminology. They apply the terms “adhesive energy”, “adhesive work”, “adhesive capacity” as these terms were undistinguishable. The authors should stick to the term “energy” throughout the manuscript whenever referring to the binding energies defined in section 2.5.

9. The sentence “Thus, it is necessary to adopt MD method to simulate the variations of adhesion work of asphalt-aggregate interface system in different environments” in lines 274-276 should be rephrased.

10. The asphalte-aggregate binding energy displayed in Table 2 are 2 orders of grandeur larger than the values reported by Sun and Wang in their recent work on Appl. Surf. Sci. 2020, 510, 145435. The authors should either revise their values or report why such a difference, even though almost the same methodology was employed in this work.

Finally, there are numerous terms that are mistakenly used, such as “hydrophobic property” instead of “hydrophobic nature” in line 177, “powerful” instead of “strong” in line 197, “delayed” in line 242 and “postponed” in line 248 instead of “shifted”. The authors should deeply grammatically revise the manuscript.

Author Response

Dear Ms. Evelyn Chen and Reviewers,

Thanks very much for taking your time to review this manuscript. I really appreciate all your comments and suggestions! Please find my itemized responses in below and my revisions in the re-submitted files.

Sincerely,

Wentian Cui

Response to Reviewer 2 Comments:

Point 1: Still on a conceptual level, the authors did not precise if their model had an overall charge. Besides, the representation of the water-modified surfaces, identified in the asphalt/calcite aggregates with the suffix “-AW”, was rather confusing, with little detailing of the specific modification with the introduction of OH- and HCO3- ions. This “adsorbed water” notation was indeed misleading, since it induces the reader to think there is actually water adsorbed at the interface when it is just a surface modification caused by the presence of water. I suggest changing the “adsorbed water (AW)” notation to another less confusing term, e.g. “modified surface (MS)” or “hydrated surface (HS)”, to avoid this ambiguity. This applies to the whole of the manuscript, the term “adsorbed water” should be used only for the actual interfacial water simulated in the “aggregate-water-asphalt” models.

Response 1: Thank you for your suggestions. The models we used was conducted molecular dynamic simulations under COMPASS force field. COMPASS force field parameters were from ab initio methods, and the charge between atoms comes from the fitting of quantum chemistry data (Line 113-115, Page 3-4). We are very approved of your suggestion. The description of the asphalt-calcite aggregates with the suffix “-AW” is imprecise, which can make readers confused. Therefore, we have been added “hydrated surface” (Line 173-174, Page 6), and “hydrated surface (HS)” has been replaced “AW” in the whole manuscript. We have made further introduction to the models (Line 179-182, Page 6). Meanwhile, the term “adsorbed water” should be used in “Results and discussion” to describe the effect of moisture at asphalt-aggregate interface. Thank you again for your valuable advice.

Point 2: Regarding the construction of the asphalt-aggregate models with adsorbed water at the interface, the authors did not precise the amount of water used in their models and neither justified such amount with experimental data or other previously simulation results.

Response 2: We are very sorry for not added the reference of “300 water molecules”. It has been added (Line 187, Page 6).

Point 3: The authors also did not precise their choice of charges. Even though the COMPASS forcefield, as implemented in the Materials Studio suite has built-in charges, this should be explicitly cited in the text.

Response 3: Thank you for your suggestion. We should precisely explain the source of charges of field force. We have been added it (Line 114-115, Page 3-4).

Point 4: The authors should precise in further details the debonding energy in section 2.5.

Response 4: Thank you for your suggestions. We have been added more detailed information about debonding energy in section 2.5 (Line 226-230, Page 9).

Point 5: In their results and discussion section, the authors draw some conclusions that need further validation, such as:

- In line 192-193, “This was due to electrostatic attractions between hydrogen bonds on the surface of weakly alkaline minerals and the polarity of resin”. A RDF between the polar groups of the resin and the proximal aggregate atoms should be presented to support this affirmation.

- In line 211-212, “This was not beneficial This was not beneficial for the interaction of calcite and asphaltene and the adhesive energy of asphalt-calcite interface was attenuated.”. A mention to the calculated adhesive energies in section 3.2 should have been stated.

- In line 291 “(…) and affected the service life of asphalt mixtures.” A reference should be mentioned to relate this cause-effect relationship.

Response 5: Thank you for your suggestions. We should adopt RDF data to support that the hydrogen bond on aggregate surface is electrostatically attracted to the resin with powerful polarity through the explanation of the increase of g (r) value (Line 245, Page9). We are sorry that the meaning of this sentence is not clearly stated. The sentence should be described that the water can attenuate the adhesive ability between asphalt and calcite, not the “adhesion energy” which was analyzed in 3.2. We have been corrected it (Line 266, Page 9). We have added references to the cause-effect relationship (Line 349, Page 14). Thank you again for your significant suggestions.

Point 6: The choice of color for the RDFs is very confusing, specially the yellow curves in a white background. It is quite difficult to distinguish some of these curves in some cases.

Response 6: We are very sorry for the choice of RDF curves color. We have been modified it (Line 272-274, Page 10; Line 278-280, Page 11).

Point 7: In lines 230 and 231, the term “Dynamic trajectory was employed to evaluate relative concentration along X direction” was confusing. The authors tracked the relative concentration along the vertical direction and should not mention “X direction” but “X axis”.

Response 7: We are very sorry for the unclear description, it should be “X axis”. We have been corrected it (Line 286, Page 11).

Point 8: The authors should be homogeneous on their terminology. They apply the terms “adhesive energy”, “adhesive work”, “adhesive capacity” as these terms were undistinguishable. The authors should stick to the term “energy” throughout the manuscript whenever referring to the binding energies defined in section 2.5.

Response 8: We are very appreciated for your suggestion. The terminology “adhesion energy” should be consistent. We have been modified it (such as Line 19,20,23, Page 1.etc) all the paper.

Point 9: The sentence “Thus, it is necessary to adopt MD method to simulate the variations of adhesion work of asphalt-aggregate interface system in different environments” in lines 274-276 should be rephrased.

Response 9: Thank you for your suggestion. The sentence has been rephrased (Line 330-332, Page 14).

Point 10: The asphalt-aggregate binding energy displayed in Table 2 are 2 orders of grandeur larger than the values reported by Sun and Wang in their recent work on Appl. Surf. Sci. 2020, 510, 145435. The authors should either revise their values or report why such a difference, even though almost the same methodology was employed in this work. Finally, there are numerous terms that are mistakenly used, such as “hydrophobic property” instead of “hydrophobic nature” in line 177, “powerful” instead of “strong” in line 197, “delayed” in line 242 and “postponed” in line 248 instead of “shifted”. The authors should deeply grammatically revise the manuscript.

Response 10: Thank you for your suggestion. We have been carefully studied the article (Sun, W.; Wang, H. Moisture effect on nanostructure and adhesion energy of asphalt on aggregate surface: A molecular dynamics study, Appl. Surf. Sci. 2020, 510, 145435). In this paper, Figure 7 and 9 show the adhesion energy of the asphalt-aggregate interface system, which is the same order of magnitude with our Figure 10. Meanwhile, the data of debonding energy similar to our Table 3 could not found in the paper. In the article “Liu, J.Z.; Yu, B.; Hong, Q.Z. Molecular dynamics simulation of distribution and adhesion of asphalt components on steel slag, Constr. Build. Mater. 2020, 255, 119332”, the debonding energy data of Table 3 is the same order of magnitude with our Table 3.

We are very sorry for the unclear terms. We have been modified them (Line 226, Page 9; Line 251, Page 9; Line 297, Page 12; Line 303, Page 12), and revised the whole manuscript. Thanks again for your advice.

Round 2

Reviewer 2 Report

Considering the authors abided to my previous suggestions and reshaped and revised the article almost in its totality, I suggest the article to be published.